# Substitution of Mineral Fertilizer with Organic Fertilizer in Maize Systems: A Meta-Analysis of Reduced Nitrogen and Carbon Emissions

**Zhibiao Wei [1,2]**, **Hao Ying [1]**, **Xiaowei Guo [1]**, **Minghao Zhuang [1]**, **Zhenling Cui [1,*]** **and Fusuo Zhang [1]**

[1]   Center for Resources, Environment and Food Security, China Agricultural University, Beijing 100193, China; zhibiao.wei@wur.nl (Z.W.); yingrl@163.com (H.Y.); wxiaoguo17@163.com (X.G.); zhuangminghao3@163.com (M.Z.); zhangfs@cau.edu.cn (F.Z.)

[2]   Department of Soil Quality, Wageningen University & Research, P.O. Box 47, 6700 AA Wageningen, The Netherlands

[*]   Correspondence: cuizl@cau.edu.cn; Tel.: +86-010-6273-3454

**Abstract:** Organic fertilizer is an effective substitute for mineral fertilizer that improves crop yield and is environmentally friendly. However, the effects of substitution often vary due to complicated interactions among the organic fertilizer substitution rate (Rs), total nutrient supply, and type of cropping system used. We performed a meta-analysis of 133 maize studies, conducted worldwide, to assess maize yield and environmental performance with substitution of mineral fertilizer with organic fertilizer. At an equivalent nitrogen (N) rate, substituting mineral fertilizer with organic fertilizer increased maize yield by 4.22%, reduced $NH_3$ volatilization by 64.8%, reduced N leaching and runoff by 26.9%, and increased $CO_2$ emissions by 26.8%; however, it had no significant effect on $N_2O$ or $CH_4$ emissions. Moreover, substitution with organic fertilizer increased the soil organic carbon sequestration rate by 925 kg C ha$^{-1}$ yr$^{-1}$ and decreased the global warming potential by 116 kg $CO_2$ eq ha$^{-1}$ compared with mineral fertilizer treatment. The net global warming potential after organic fertilizer substitution was −3507 kg $CO_2$ eq ha$^{-1}$, indicating a net carbon sink. Furthermore, the effect of organic fertilizer substitution varied with the fertilization rate, Rs, and treatment duration. Maize yield and nitrogen use efficiency tended to increase with increasing N application rate following substitution of mineral fertilizer with organic fertilizer. Full substitution reduced N losses more than partial substitution. Further analysis revealed that the yield-optimal Rs for organic N in maize production was 40–60%. Moreover, maize yield and nitrogen use efficiency were further increased after long-term (≥ 3 years) combined use of organic and mineral fertilizers. These findings suggest that rational use of organic and mineral fertilizers improves maize productivity, increases soil organic carbon sequestration, and reduces N and C losses.

**Keywords:** organic fertilizer substitution; fertilization rate; maize productivity; N and C emissions; net global warming potential

## 1. Introduction

Fertilizer has played a key role in global food safety over recent decades, which is necessary to meet the needs of the increasing world population [1]. However, overuse of mineral fertilizer introduces substantial reactive nitrogen (Nr) emissions into the environment, resulting in adverse effects such as air pollution [2,3], soil degradation and acidification [4], and water eutrophication [5,6]. The emitted Nr to air is a major precursor to form fine particular matter with the aerodynamic diameter less than 2.5 μm (PM2.5), which is a threat to human health [5]. The nitrate discharged to water

could cause biodiversity losses [5,6]. Global annual production of livestock manure nitrogen (N) has reached nearly 100 Tg N [7]. Recycling of manure and other organic materials into the field could potentially create a "win-win" situation, by reducing mineral fertilizer input while simultaneously addressing sanitation problems [8,9]. However, the contaminants in it, such as antibiotics or heavy metals, could limit the widespread use of household and animal waste [10,11]. Many researchers have explored the effect of substituting mineral fertilizer with organic fertilizer. Organic fertilizer application is also regarded as an efficient way to increase soil organic carbon (SOC) [12], which not only enhances crop production [13], but also acts as a conditioner to improve soil resilience and health, reduce C ($CO_2$) and N ($N_2O$, nitrate) emissions to the environment and increase water retention [13,14]. A meta-analysis of studies performed in Africa demonstrated that combined use of organic and mineral fertilizers increased crop yield, but the substitution rates (Rs) used were not specified [15]. A study carried out in China analyzed the effect of organic fertilizer Rs on crop productivity and Nr losses and found that the yield-optimal Rs for manure N was 50–75% [16]. However, the usefulness of previous studies is limited by suboptimal N rates, and variation in application timing and duration [12,17]. Comparison of the agronomic and environmental effects of organic fertilizer substitution is difficult because the optimal N supply rate is highly variable among years and sites.

Substitution of mineral fertilizer with organic fertilizer has multiple effects on crop production, environmental emissions, and SOC sequestration. Most studies only evaluated one aspect, such as crop yield [18], nitrogen use efficiency (NUE) [19], SOC [12], or environmental emissions [17]. Xia et al. (2017) evaluated the net global warming potential (NGWP) associated with manure substitution; they found that substituting manure for mineral fertilizer increased the carbon (C) sink in upland fields [16], but they did not consider the indirect effects of $NH_3$ emissions and runoff/leaching on NGWP. Moreover, most reviews of research on organic fertilizer substitution considered all crops in combination, thus providing data with limited utility for assessing the crop-specific effects of organic fertilizer substitution on yield and environmental emissions [17,18].

In this study, we performed a meta-analysis of studies conducted worldwide to quantitatively assess the effects of substituting organic fertilizers for mineral fertilizers on maize production, N and C emissions, and the soil organic carbon sequestration rate (SOCSR). We also evaluated the global warming potential (GWP) and NGWP, considering the indirect effects of $NH_3$ emissions and runoff/leaching. The responses of these variables to organic fertilizer substitution were evaluated according to the fertilization rate (low, optimal, or high), Rs, and treatment duration. The underlying causes of the different responses of these variables to substitution of chemical with organic fertilizer are discussed.

## 2. Materials and Methods

### 2.1. Data Collection

Peer-reviewed articles were searched for to evaluate the effects of substitution of mineral fertilizer with organic fertilizer on maize yield, N and C losses, and SOC sequestration. Studies published before March 2020 were searched for in the Microsoft Academic (https://academic.microsoft.com/home), Google Scholar (https://scholar.google.com/), Baidu Scholar (http://xueshu.baidu.com/), and China National Knowledge Infrastructure (http://www.cnki.net/) databases. Search terms related to maize production, organic fertilizer substitution, manure application, and N and C emissions were combined.

The studies included in our analysis satisfied the following criteria: (1) Studies focused on maize production with substitution of mineral fertilizer with organic fertilizer, including animal manure (47%), compost (37%), commercial organic fertilizer (e.g., industrially processed, standardized poultry or livestock manure; 9%), digestate (5%), slurry (2%); (2) The amounts of applied organic material and the N content were clearly specified; (3) The mineral fertilizer treatment and "substitution treatments" (partial or full substitution of chemical with organic fertilizer) had identical total N rates, and phosphorus (P) and potassium (K) inputs were not major factors limiting maize growth; (4) The N application rates for each treatment were reported to allow calculation of the Rs, defined as organic



N input/total N applied; (5) Field or lysimeter study, with articles reporting data from the same experiment excluded; and (6) Experimental duration of at least 3 years with respect to the effects of organic fertilizer on soil properties (e.g., SOC) [20]. In total, 133 published articles performed worldwide were included in the analysis (see supporting information for details).

## 2.2. Evaluated Variables and Data Treatment

Eight dependent/response variables were evaluated to determine the effects of substitution of mineral fertilizer with organic fertilizer, categorized as follows: (1) maize productivity: maize yield and NUE; (2) N losses: $NH_3$ emissions, $N_2O$ emissions, and N runoff/leaching; and (3) C emissions ($CO_2$ and $CH_4$) and SOCSR. NUE, based on the N recovery rate, was calculated using the following equation:

$$NUE = (U_f - U_0)/F \tag{1}$$

where $U_f$ and $U_0$ are the aboveground N uptake of maize in plots with and without fertilization, respectively; F indicates total fertilizer N input.

For studies that did not report SOCSR directly, it was calculated using the following equation [16]:

$$SOCSR\ (kg\ C\ ha^{-1}\ yr^{-1}) = (SOC_f \times \rho_f - SOC_0 \times \rho_0) \times H \times 100/T \tag{2}$$

where $SOC_f$ and $SOC_0$ are the soil organic content (kg C $t^{-1}$) in plots with and without fertilization, respectively; $\rho_f$ and $\rho_0$ are the soil bulk density (t $m^{-3}$) for the fertilizer and control treatments, respectively; H is the sampling depth (cm); and T is the treatment duration. In studies with missing $\rho$-values, they were estimated using the following equation [16]:

$$\rho = -0.0048 \ln SOC + 1.377 \tag{3}$$

The effects of substitution of mineral fertilizer with organic fertilizer were evaluated according to fertilization rate, Rs, and treatment duration. The optimal total fertilizer and organic fertilizer Rs were determined in terms of their effects on maize yield and NUE. The N fertilization rate for maize production was categorized as low, optimal, or high, where these categories were region-specific: the optimal N rates for Africa [20,21], Asia [22], and other regions [23,24] were 50–80, 150–210, and 180–250 kg N $ha^{-1}$, respectively. The Rs, defined as organic N input/total N input, was divided into four categories (0 < Rs ≤ 40, 40 < Rs ≤ 60, 60 < Rs < 100, and Rs = 100) when analyzing maize productivity and N emissions. Two Rs categories, i.e., full (Rs = 100) and partial (0 < Rs < 100), were used when analyzing C emissions and SOCRS, as these data were limited. Treatment duration was classified as short (< 3 years) or long (≥ 3 years) for all dependent variables.

## 2.3. Meta-Analysis

A standardized effect size, which reflects the magnitude of the substitution treatment effect compared with the control, was calculated for all studies to allow robust statistical comparison, given that the studies reported results based on different variables [17]. To derive this standardized effect size, the natural log of the response ratio (lnR) was calculated [25], except for $CH_4$ and SOCSR, as follows:

$$\ln RR = \ln(X_o/X_m) \tag{4}$$

where $X_o$ and $X_m$ are the mean values of variable X (e.g., maize yield, NUE, N emissions, C emissions) in the organic and mineral fertilizer treatments, respectively. Log transformation of the response ratio was used to stabilize the variance. Results were exponentially back-transformed and converted to percentage change values relative to the control treatment [(RR − 1) * 100]. Negative and positive percentage changes indicate a decrease or increase, respectively, in the corresponding response variable

due to organic fertilizer substitution. The 95% confidence intervals (CIs) not overlapping zero denote significant differences.

The $CH_4$ emissions and SOCSR values can be positive or negative; thus, Equation (4) is undefined. The effect sizes for these two variables were calculated based on the mean difference ($RR_2$):

$$RR_2 = X_o - X_m \tag{5}$$

Effect sizes were weighted by the inverse of the sampling variance [26,27]. For studies where neither the standard deviation (SD) nor standard error (SE) were reported, the approach of Bracken [28] was applied to estimate SD using in the "metagear" R (ver. 3.6.1; R Development Core Team, Vienna, Austria) package. A random-effect model was employed for the meta-analysis, generated using the "metafor" R package. Maize production and N and C emissions databases were generated based on the study data using Excel 2010 software (Microsoft Corp., Redmond, WA, USA).

*2.4. Net Global Warming Potential*

The GWP of organic fertilizer substitution for mineral fertilizer was calculated based on changes in emissions of $CH_4$, $N_2O$, and $NH_3$, and leaching/runoff:

$$GWP \text{ (kg } CO_2 \text{ eq ha}^{-1}) = (NH_3\text{-}N_{change} \times 0.01 + NO_3^-\text{-}N_{change} \times 0.0075 + N_2O\text{-}N_{change}) \times$$
$$298 \times 44/28 + CH_4\text{-}C_{change} \times 25 \times 16/12 \tag{6}$$

where $NH_3\text{-}N_{change}$, $NO_3^-\text{-}N_{change}$, $N_2O\text{-}N_{change}$, and $CH_4\text{-}C_{change}$ are the emission changes (kg ha$^{-1}$) induced by organic fertilizer substitution. The values 298 and 25 represent the GWP of $N_2O$ and $CH_4$ in units of $CO_2$ equivalents over a 100-year period according to the Intergovernmental Panel on Climate Change (IPCC) Fifth Assessment Report [29]. The values 0.01 and 0.0075 represent the indirect $N_2O$ emissions from volatilized $NH_3$-N and leached $NO_3^-$-N, according to IPCC methodology [30]. The values of 44/28 and 16/12 indicate mass conversion factors of N to $N_2O$, and C to $CH_4$.

The NGWP was calculated using the following equation:

$$NGWP = GWP - SOCSR_{change} \times 44/12 \tag{7}$$

where $SOCSR_{change}$ is the SOCSR change induced by substituting mineral fertilizer with organic fertilizer. A negative value for NGWP indicates a net C sink, whereas a positive value represents a net C source.

The changes in $CH_4$, $N_2O$, and $NH_3$ emissions, as well as leaching/runoff and SOCSR, were calculated separately for the full and partial substitution groups, to determine the GWP and NGWP by organic fertilizer Rs.

## 3. Results

*3.1. Effects of Organic Fertilizer Substitution on N and C Emissions by Fertilization Rate*

Substitution of mineral fertilizer with organic fertilizer significantly increased maize yield (by 4.23%) but had no significant effect on NUE (Figure 1). We found that maize yield and NUE tended to increase with increasing N fertilization rate. The low fertilization rate had a negative impact on NUE (−17.3%) and did not significantly enhance maize yield. The optimal fertilization rate significantly increased maize yield, but compromised NUE to some extent. The high fertilization rate significantly increased maize yield (4.83%) and NUE (8.77%).

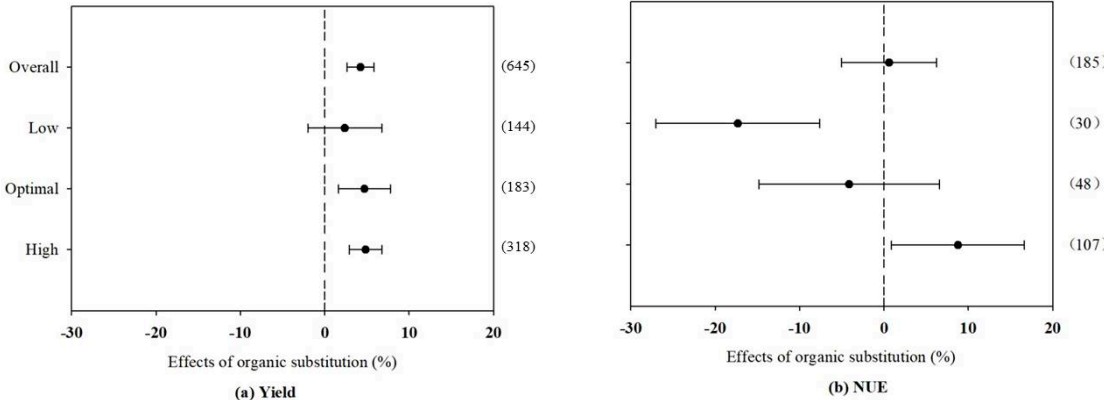

**Figure 1.** Effects of substitution of mineral fertilizer with organic fertilizer (under three fertilization rates; see main text for definitions of the low, optimal, and high fertilization rates) on maize yield (**a**) and nitrogen use efficiency (NUE) (**b**). Error bars represent 95% confidence intervals. Numbers in parentheses indicate the numbers of observations.

Organic fertilizer substitution significantly decreased $NH_3$ emissions (by 64.8%) and N leaching and runoff (by 26.8%), whereas $N_2O$ emissions were decreased non-significantly (by 12.7%). Organic fertilizer substitution decreased N losses at all fertilization rates, except for $N_2O$ at the low rate, which was significantly increased (by 33.1%). The decrease in $NH_3$ and $N_2O$ emissions tended to increase with increasing N fertilization rate. The maximum decrease in N runoff and leaching (by 45.5%) was achieved at the optimal fertilization rate. At the high fertilization rate, all three types of N loss ($NH_3$, $N_2O$, and runoff and leaching) were significantly decreased (by 71.7%, 37.1%, and 22.1%, respectively; Figure 2).

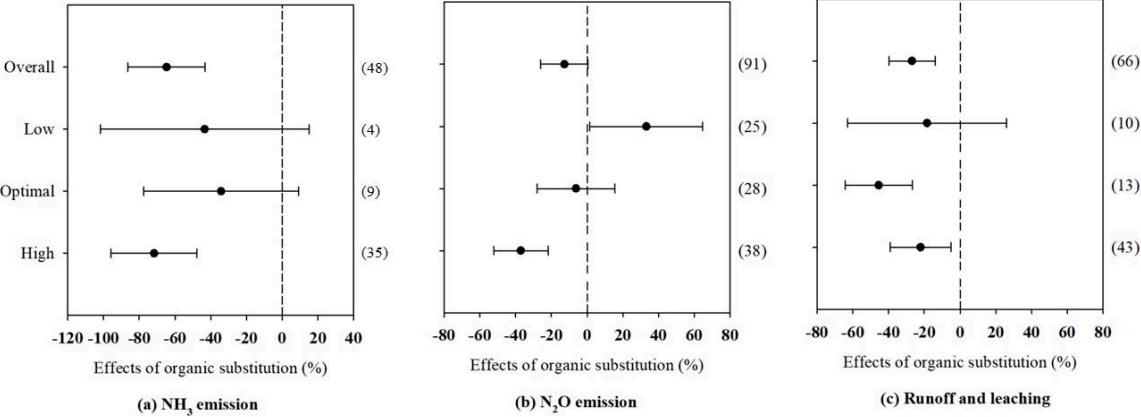

**Figure 2.** Effects of organic fertilizer substitution on $NH_3$ emissions (**a**), $N_2O$ emissions (**b**), and runoff and leaching (**c**) under three fertilization rates (see main text for definitions of low, optimal, and high fertilization rates). Error bars represent 95% confidence intervals. Numbers in parentheses indicate the numbers of observations.

Organic fertilizer substitution significantly increased $CO_2$ emissions (by 26.8%) and the SOCSR (by 925 kg C ha$^{-1}$ yr$^{-1}$) but had no effect on $CH_4$ emissions. $CO_2$ emissions showed a decreasing trend as the rate of N fertilization increased, whereas the SOCSR showed an increasing trend as the fertilization rate increased. $CH_4$ emissions were not significantly affected by organic fertilizer substitution at any of the three fertilization rates (Figure 3).

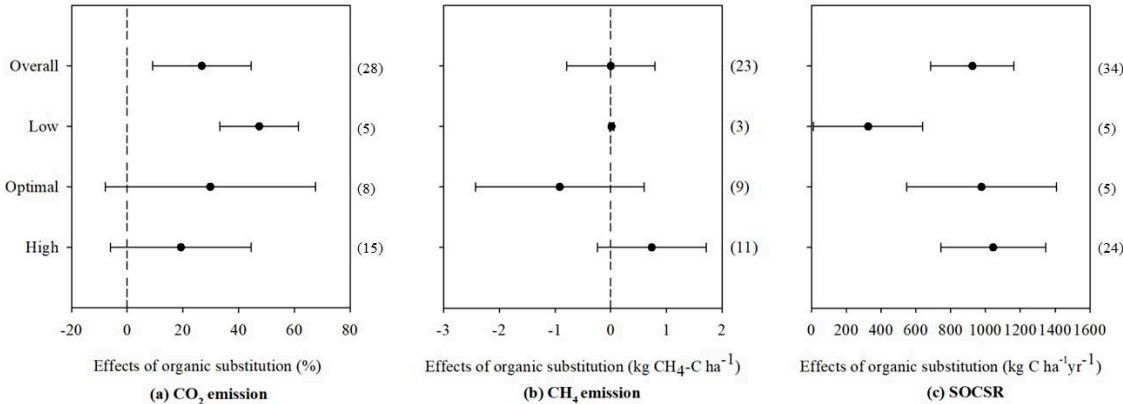

**Figure 3.** Effects of organic fertilizer substitution on $CO_2$ emissions (**a**), $CH_4$ emissions (**b**), and SOCSR (**c**) under three fertilization rates (see main text for definitions of the low, optimal, and high fertilization rates). SOCSR indicates soil organic carbon sequestration rate. Error bars represent 95% confidence intervals. Numbers in parentheses indicate the numbers of observations.

*3.2. Effects of Organic Fertilizer Substitution on N and C Emissions at Different Substitution Rates*

The yield-optimal Rs for organic fertilizer was 40–60%, and gave rise to a statistically significant increase in maize yield (11.5%). For Rs values between 60% and 100%, the yield increases were not significant. Full substitution of chemical with organic fertilizer (Rs = 100) decreased the maize yield to some extent. NUE decreased gradually from 17.3% to −20.4% as the Rs increased. NUE was increased significantly at the low Rs, but was decreased significantly by full substitution (Figure 4).

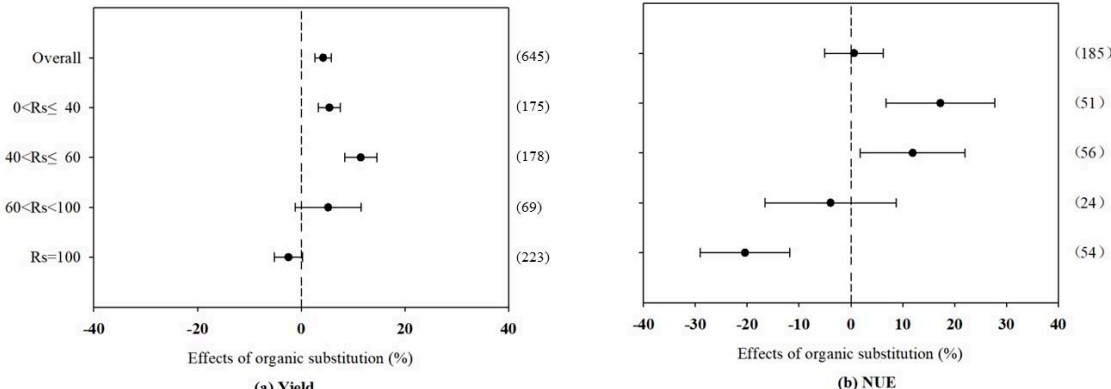

**Figure 4.** Effects of four rates of organic fertilizer substitution on maize yield (**a**) and nitrogen use efficiency (NUE) (**b**). Rs, substitution rate, defined as organic nitrogen (N) input/total N applied (%). Error bars represent 95% confidence intervals. Numbers in parentheses indicate the numbers of observations.

At the optimal fertilization rate, the yield-optimal Rs was between 60% and 100%, whereas with the low and high fertilization rates the optimal Rs was between 40% and 60%. NUE tended to decrease as the Rs increased at the low and high fertilization rates, whereas it did not change consistently at the optimal fertilization rate. Maize yield and NUE were both low at the full Rs for all three fertilization rates (Figure 5).

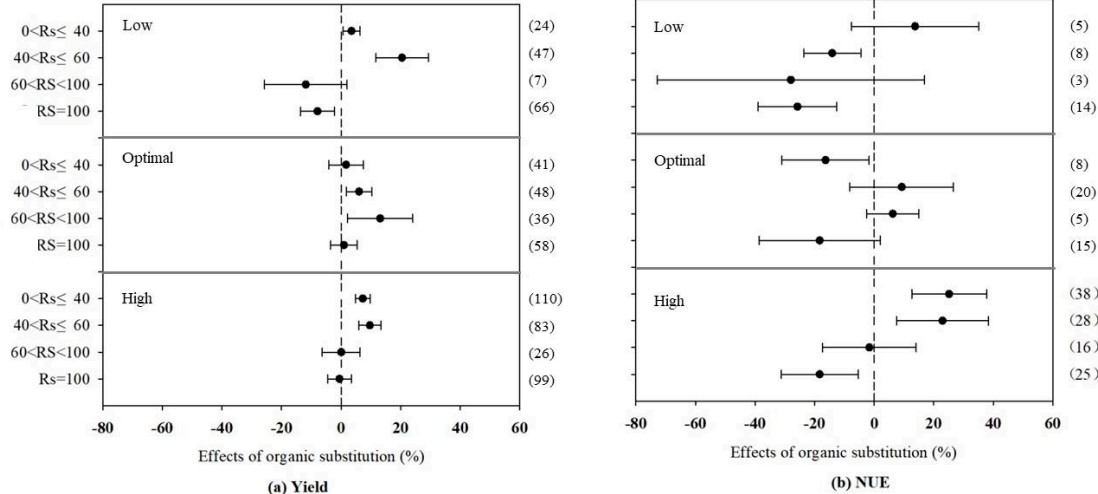

**Figure 5.** Effects of combined organic fertilizer substitution on maize yield (**a**) and nitrogen use efficiency (NUE) (**b**). Rs, substitution rate, defined as organic nitrogen (N) input/total N applied (%). See main text for definitions of low, optimal, and high fertilization rates. Error bars represent 95% confidence intervals. Numbers in parentheses indicate the numbers of observations.

Partial substitution (0 < Rs < 100) of organic with mineral fertilizer did not significantly affect N losses, except for $N_2O$ at a partial Rs of 0–40%, and runoff and leaching at a rate of 40–60%. $N_2O$ emissions were significantly increased (by 23.7%) when the Rs was between 0% and 40%. Runoff and leaching were significantly decreased (by 41.9%) when the Rs was between 40% and 60% (Figure 6). Full substitution of organic with mineral fertilizer significantly decreased all types of N loss (93.0% for $NH_3$, 25.0% for $N_2O$, and 50.0% for runoff and leaching).

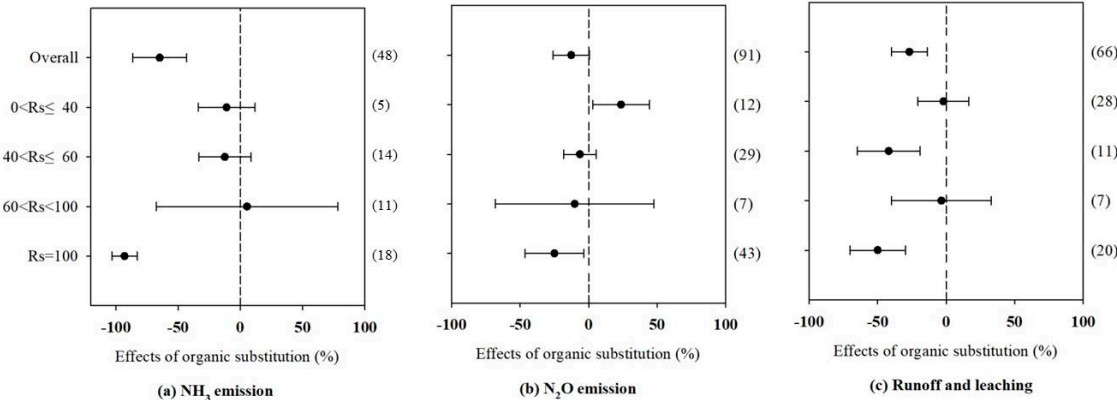

**Figure 6.** Effects of organic fertilizer substitution on $NH_3$ emissions (**a**), $N_2O$ emissions (**b**), and runoff and leaching (**c**) under four substitution rates. Rs, substitution rate, defined as organic nitrogen (N) input/total N applied (%). Error bars represent 95% confidence intervals. Numbers in parentheses indicate the numbers of observations.

The variation in C emissions by Rs was large, probably due to the limited amount of available data. Partial substitution of mineral fertilizer with organic fertilizer did not significantly affect $CO_2$ emissions, while full substitution significantly increased $CO_2$ emissions (by 41.9%). $CH_4$ emissions were not significantly affected by the Rs. The SOCSR under partial substitution was 968 kg C ha$^{-1}$ yr$^{-1}$, which was slightly higher than that under full substitution (817 kg C ha$^{-1}$ yr$^{-1}$; Figure 7).

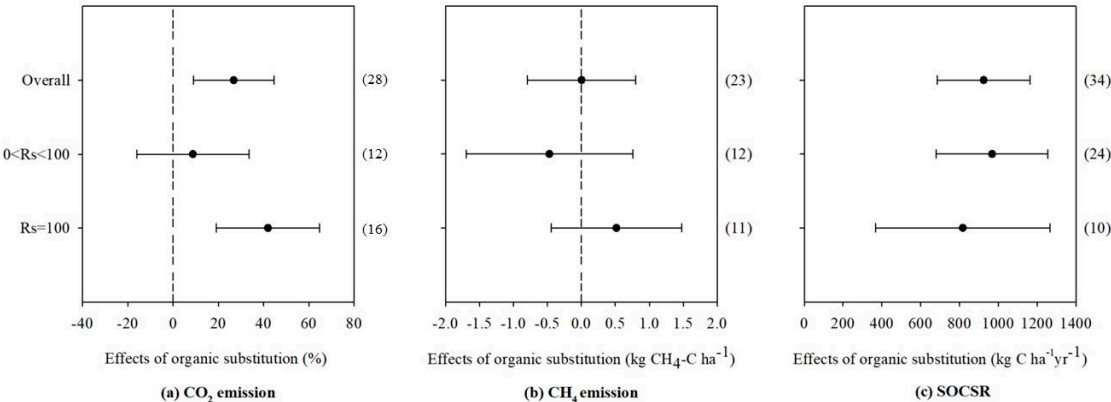

**Figure 7.** Effects of organic fertilizer substitution on $CO_2$ emissions (**a**), $CH_4$ emissions (**b**), and the soil organic carbon sequestration rate (SOCSR) (**c**) for maize production under four substitution rates. Rs, substitution rate, defined as organic nitrogen (N) input/total N applied (%). Error bars represent 95% confidence intervals. Numbers in parentheses indicate the numbers of observations.

### 3.3. Effects of Organic Fertilizer Substitution on N and C Emissions by Treatment Duration

Long-term application (≥3 years) of organic fertilizer significantly increased maize yield (by 11.5%) without compromising NUE, whereas short-term application (<3 years) did not affect maize yield or NUE. $NH_3$ and $N_2O$ emissions were promoted by long-term organic fertilizer substitution in comparison with short-term application. Runoff and leaching were significantly decreased, by 47%, following long-term organic fertilizer substitution; this was significantly greater than the 11.2% reduction under short-term treatments. $CH_4$ emissions under long-term treatments decreased slightly, by 0.63 kg $CH_4$-C ha$^{-1}$, but increased by 0.69 kg $CH_4$-C ha$^{-1}$ over the short term. $CO_2$ emissions were increased by 54.3% under long-term organic fertilizer substitution but did not change significantly over the short term (Figure 8).

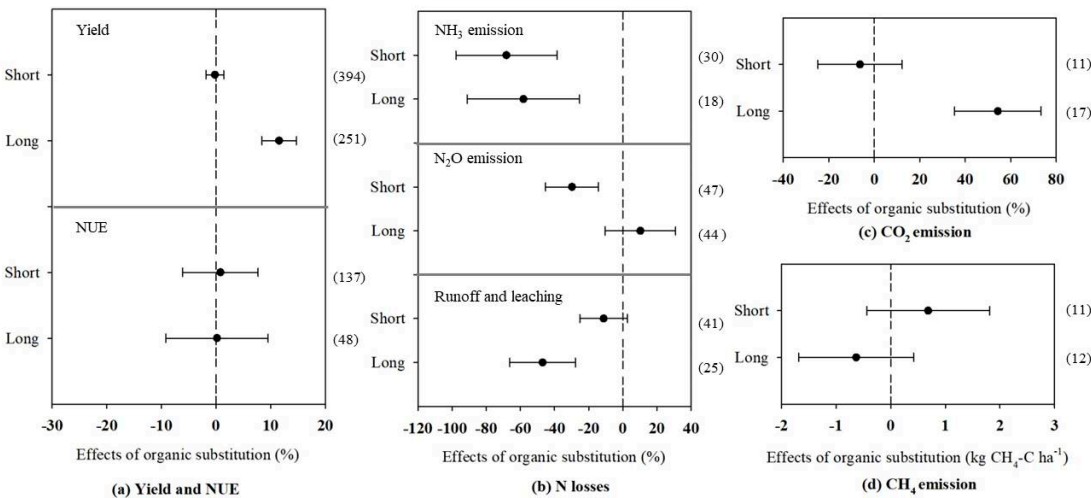

**Figure 8.** Effects of organic fertilizer substitution on maize yield and nitrogen use efficiency (NUE) (**a**), nitrogen (N) losses (**b**), $CO_2$ emissions (**c**), and $CH_4$ emissions (**d**) under two treatment durations (short: <3 years, long: ≥3 years). N losses include $NH_3$ emissions, $N_2O$ emissions, and runoff and leaching. Error bars represent 95% confidence intervals. Numbers in parentheses indicate the numbers of observations.

### 3.4. Effects of Organic Fertilizer Substitution on GWP and NGWP

Organic fertilizer substitution reduced greenhouse gas emissions (Table 1). The overall GWP decreased by 116 kg $CO_2$ eq ha$^{-1}$ with organic fertilizer substitution compared with chemical

fertilization. Full replacement of mineral fertilizer with organic fertilizer reduced greenhouse gas emissions (203 kg $CO_2$ eq ha$^{-1}$) to a greater extent than partial substitution (67.2 kg $CO_2$ eq ha$^{-1}$), because volatilized $NH_3$ and leached $NO_3^-$ were reduced by full substitution. However, when considering NGWP, more C was sequestered with partial substitution (3617 kg $CO_2$ eq ha$^{-1}$) than with full substitution (3200 kg $CO_2$ eq ha$^{-1}$), because the SOCSR for partial substitution was larger than that for full substitution. The higher rate of soil C sequestration with partial substitution outweighed its adverse effect on greenhouse gas emissions compared with full substitution. The overall NGWP with organic fertilizer substitution was −3507 kg $CO_2$ eq ha$^{-1}$, indicating that a net C sink was produced by substitution of chemical with organic fertilizer.

**Table 1.** Net global warming potential parameter values according to organic fertilizer substitution amount.

| Parameter | Overall | Full Substitution | Partial Substitution |
|---|---|---|---|
| $NH_3$ (kg N ha$^{-1}$) | −14.6 | −31.6 | −4.13 |
| $N_2O$ (kg N ha$^{-1}$) | −0.06 | −0.05 | −0.05 |
| Leaching/runoff (kg N ha$^{-1}$) | −5.63 | −13.5 | −2.15 |
| $CH_4$ (kg C ha$^{-1}$) | 0.00 | 0.52 | −0.47 |
| SOCSR (kg C ha$^{-1}$ yr$^{-1}$) | 925 | 817 | 968 |
| GWP (kg $CO_2$ eq ha$^{-1}$) | −116 | −203 | −67.2 |
| NGWP (kg $CO_2$ eq ha$^{-1}$) | −3507 | −3200 | −3617 |

SOCSR, soil organic carbon sequestration rate; GWP, global warming potential; NGWP, net global warming potential; Negative NGWP values indicate a net carbon sink as a result of organic fertilizer substitution.

## 4. Discussion

### 4.1. Maize Productivity and Soil Conditions

Substitution of mineral fertilizer with organic fertilizer significantly increased maize yield, probably by improving soil properties, and increasing SOC and total N (Table 2). The SOC content increased from 10.37 g kg$^{-1}$ for mineral fertilizer treatment to 13.28 g kg$^{-1}$ for organic fertilizer treatment, which could explain the positive SOCSR following substitution of chemical with organic fertilizer. The soil TN content increased from 1.1 to 1.29 g kg$^{-1}$ following organic fertilizer substitution (Table 2). Full substitution of chemical with organic fertilizer decreased maize yield to some extent (Figures 4 and 5), mainly because the in-season utilization rate is lower for organic N than for chemical N [31]; this could also explain the low NUE following organic fertilizer substitution.

Organic fertilizer substitution also reduced soil acidification caused by mineral fertilizer application [4]. The soil pH increased from 6.47 with mineral fertilizer treatment to 6.77 with organic fertilizer substitution treatment according to our literature review; nevertheless, the soil pH still decreased after long-term organic inputs in calcareous soil (Table 2). This may be because the soil pHs reported in the included articles was high (7.68), while the pH values of some organic fertilizers, especially those based on pig and poultry manure, can be less than 6.5 [32]. As the abundant humic acid in some organic materials decomposes, it releases $H^+$ into the soil [33], which in turn decreases the pH of alkaline soil. We noticed that, conversely, organic fertilizer substitution increased the pH of acid soil [34]. Organic inputs therefore improve soil pH resilience. Soil aggregation [35] and water-holding capacity [36,37] were also improved after long-term organic substitution. Improved water-holding capacity is extremely important for rain-fed, water-starved maize planting systems.

**Table 2.** Effect of chemical and organic fertilizers on soil properties.

| Soil Property | Control | | | Mineral Fertilizer | | | Organic Fertilizer | | |
| --- | --- | --- | --- | --- | --- | --- | --- | --- | --- |
| | Mean | SD | n | Mean | SD | n | Mean | SD | n |
| SOC (g kg$^{-1}$) [20,34–46] | 9.69 | 2.16 | 22 | 10.37 | 2.87 | 30 | 13.28 | 4.38 | 30 |
| TN (g kg$^{-1}$) [34,35,37–39,41,42,44,46] | 1.09 | 0.15 | 14 | 1.10 | 0.26 | 19 | 1.29 | 0.36 | 19 |
| pH [34,35,37–39,41,44,46] | 7.68 | 1.12 | 13 | 6.47 | 1.66 | 18 | 6.77 | 1.27 | 18 |
| BD (g cm$^{-3}$) [36,41,46] | 1.35 | 0.09 | 8 | 1.32 | 0.10 | 11 | 1.26 | 0.05 | 11 |

All data were from long-term experiments (≥3 years). SOC, soil organic carbon; TN, soil total nitrogen; BD, soil bulk density; SD, standard deviation.

*4.2. N and C emissions*

Organic fertilizer substitution tended to decrease all types of reactive N loss (Figures 2 and 5). This may have been due to reduced availability of Nr (the major type of N loss) with organic fertilizer substitution treatment. The increase in $CO_2$ emissions seen after organic fertilizer substitution was related to soil respiration, which was enhanced by increased C input into the soil (Table 2). The variation in $CH_4$ emissions by Rs was large, but not significant, probably due to the limited amount of available data. The complexity of the $CH_4$ production and oxidation processes could also explain the lack of significance. On one hand, the increased C inputs from organic fertilizer treatment suggest that more substrate is available for $CH_4$ formation [47]. On the other hand, organic fertilizer treatment generally leads to lower ammonium content in soil compared with mineral fertilizer treatment. Ammonium normally inhibits $CH_4$ oxidation because it increases the population of nitrifiers relative to methanotrophs, and nitrifiers oxidize $CH_4$ less efficiently than methanotrophs, such that $CH_4$ oxidizing activity changes to nitrification; therefore, lower ammonium content induces more $CH_4$ oxidation [48]. These two processes counteract each other, such that $CH_4$ emissions are not significantly affected by organic fertilizer substitution.

The overall GWP decreased by 203 kg $CO_2$ eq ha$^{-1}$ with full organic fertilizer substitution compared with chemical fertilization (Table 2). When considering the production of greenhouse gases (GHG) throughout the entire life cycle of organic and mineral fertilizers, the result could be different. The life cycle GHG emissions during mineral N fertilizer production are 8.2 kg $CO_2$ eq kg N$^{-1}$ [49]. However, life cycle GHG emissions during organic fertilizer production are highly variable, ranging from −40 to 45 kg $CO_2$ eq kg N$^{-1}$, depending on the production processes and manure types [50]. Negative value means the potential to save $CO_2$. For example, anaerobic digestion could potentially reduce $CO_2$ emission by replacing fossil fuel-based energy with biogas [50]. It is hard to conclude which one is better when comparing the entire life cycle GHG emissions of organic and mineral fertilizers. This study focuses on emissions after fertilizer application, rather than the life cycle emissions.

*4.3. SOCSR*

Two approaches are used to estimate the SOCSR; the method employed depends on the time scale and available data [47]. In short-term experiments, the SOCSR can be estimated from the difference between organic inputs and soil $CO_2$ emissions. Organic inputs include organic fertilizer and biomass remaining in the field. However, in long-term experiments, the SOCSR is generally calculated based on the inter-annual changes in SOC (Equation (2)). Our meta-analysis demonstrated that the SOCSR for partial organic fertilizer substitution treatment (968 kg C ha$^{-1}$ yr$^{-1}$) was higher than for full substitution treatment (817 kg C ha$^{-1}$ yr$^{-1}$) (Table 2). This could be explained by the significantly higher maize yield obtained with the partial substitution treatments (Figure 4), which in turn indicates a larger amount of belowground root and other biomass residue remaining in the field. Considering that the maize straw was removed from the field in most articles reviewed (data not shown), partial organic fertilizer substitution has the potential to sequester more C when straw recycling technology is used.

### 4.4. Fertilization and Substitution Rates

Our meta-analysis of 133 studies indicated that fertilization rate had a large influence on the effectiveness of organic fertilizer substitution (Figures 1–3). Organic fertilizer substitution is most effective in combination with a high fertilization rate, but we cannot conclude that a high fertilization rate is better for organic fertilizer substitution. It is known that a high mineral fertilizer rate does not increase crop yield and has a higher environmental cost [51,52]. However, a moderately higher rate of organic fertilizer substitution is associated with a greater nutrient supply that might not exceed the optimal level (because the N fertilizer replacement value of organic fertilizer is normally less than 70%) [31]. Analyzing the effect of organic fertilizer substitution without considering the fertilization rate can be misleading.

### 4.5. Limitations of This Study

Although we searched for articles without imposing any geographic limitations, most of the included studies (~80%) were carried out in China. This was because most of the organic fertilizer substitution experiments performed in Europe and America were based on available N [31], which did not meet our criterion that studies must be based on total N. Moreover, most studies performed in Africa did not report the N contents of organic materials accurately [15], which prevented verification that the total N application rates of the organic and mineral fertilizer treatments were equal. Additionally, we did not take the effects of organic treatment (compost or digestate) on organic fertilizer substitution into account [17] due to the limited amount of available data on maize production. Moreover, even though P and K inputs were not major factors limiting maize growth in the included articles, micronutrients present in organic materials, particularly boron (B) and zinc (Zn), may have contributed to the higher maize yields observed under the organic fertilizer treatments [39]. Since most studies did not report the micronutrients in organic materials, further research is needed to quantify their contribution to increased crop yield.

### 5. Conclusions

Organic fertilizer substitution significantly enhanced maize yield, reduced N losses, and promoted SOC sequestration, but also increased $CO_2$ emissions. It had no significant effect on NUE or $CH_4$ emissions during maize production. The yield-optimal Rs of organic N was 40−60%, but it varied by fertilization rate. The yield and environmental benefits of organic fertilizer substitution were higher with a high fertilization rate. Long-term organic fertilizer substitution further increased maize yield compared with short-term application. The GWP of organic fertilizer substitution was −116 kg $CO_2$ eq ha$^{-1}$, and the NGWP was −3507 kg $CO_2$ eq ha$^{-1}$, with partial substitution of mineral fertilizer with organic fertilizer sequestering more C than full substitution.

**Author Contributions:** Z.C., F.Z. and Z.W. conceived the idea. Z.C., M.Z., F.Z. supervised the project. Z.W., H.Y. and X.G. established the database. Z.W. wrote the manuscript. Z.C. and M.Z. polished the manuscript. All authors have read and agreed to the published version of the manuscript.

**Funding:** This study was funded by Qinghai Science and Technology Plan Project (2019-NK-A11), China Scholarship Council (No. 201913043).

**Acknowledgments:** We are grateful to Ellis Hoffland and Petra Hellegers for supervising the project. The first author appreciates Jie Lu for helping to tackle some technical matters on using R.

**Conflicts of Interest:** The authors declare no conflict of interest.

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
