# Peer review of "Substitution of Mineral Fertilizer with Organic Fertilizer in Maize Systems: A Meta-Analysis of Reduced Nitrogen and Carbon Emissions"

_agronomy, doi:10.3390/agronomy10081149_

Round 1
Reviewer 1 Report
- The Introduction needs to be strengthened regarding importance of soil organic matter in the environment (L45-47), negative impacts of reactive nitrogen in the environment (L40) and disadvantages of using organic fertilizers (L43).
- Since FYM, slurry, compost, digestate, and commercial organic fertilizer have differentiated impact on soil properties, especially carbon content, please specify their percentage taking into account in the meta-analysis (L82-83)
- Please specify the GWP unit in the formula (6), 44/28 and 16/12 (L 145-146)
- The net carbon balance is calculated by adding all of the carbon fluxes. It does not include nitrogen compounds.
- How did authors calculate CO2 emissions? (Fig. 3a)
- The authors should add in the discussion information on the production of greenhouse gases throughout the entire life cycle of organic and mineral fertilizers, including their production.
- All abbreviations used in tables and figures should be defined in the table notes or figure captions.
- The paper must be revised by a native English speaker.
Author Response
Q1: The Introduction needs to be strengthened regarding importance of soil organic matter in the environment (L45-47), negative impacts of reactive nitrogen in the environment (L40) and disadvantages of using organic fertilizers (L43).
A: Thanks for your suggestion. We added more information for advantages and disadvantages of using organic fertilizer in the introduction part. Increasing soil organic matter could reduce C (CO2) and N (N2O, nitrate) emissions to the environment and increase water retention (L51-52).
Apart from the listed negative impacts, the emitted Nr to air is a major precursor to form fine particular matter with the aerodynamic diameter less than 2.5 μm (PM2.5), which is a threat to human health. The nitrate discharged to water could cause biodiversity losses (L40-43).
The disadvantages of using organic fertilizers are as following: The contaminants in it, such as antibiotics or heavy metals, could limit the widespread use of household and animal waste (L46-47).
Q2: Since FYM, slurry, compost, digestate, and commercial organic fertilizer have differentiated impact on soil properties, especially carbon content, please specify their percentage taking into account in the meta-analysis (L82-83)
A: Thanks, we added them in L89-90. Animal manure (47%), compost (37%), commercial organic fertilizer (9%), digestate (5%), slurry (2%)
Q3: Please specify the GWP unit in the formula (6), 44/28 and 16/12 (L 145-146)
A: Thanks, we added the unit of GWP of kg CO2 eq ha–1 in L151 and L153.
In L158, the values of 44/28 and 16/12 are constants, indicating mass conversion factors of N to N2O, and C to CH4, respectively.
Q4: The net carbon balance is calculated by adding all of the carbon fluxes. It does not include nitrogen compounds.
A: Thank you for this suggestion. We modified our expression from “the net carbon balance” to “the net global warming potential”. The indicator of “The net global warming potential” was introduced to show the overall effects of N and C on climate change (L159-160, L162).
Q5: How did authors calculate CO2 emissions? (Fig. 3a)
A: All forms of N and C emissions, including CO2 emission, are from literature review. Please see the Materials and Methods part for details (L100-104). The CO2 emissions in the referred articles are normally measured using static chamber or micrometeorological method.
Q6: The authors should add in the discussion information on the production of greenhouse gases throughout the entire life cycle of organic and mineral fertilizers, including their production.
A: The life cycle GHG emissions during mineral N fertilizer production are 8.2 kg CO2 eq kg N–1. But life cycle GHG emissions during organic fertilizer production are highly variable, ranged from -40 to 45 kg CO2 eq kg N–1, depending on the production processes and manure types. Huge uncertainty exists to compare the entire life cycle GHG emissions of organic and mineral fertilizers. This study focuses on emissions after fertilizer application, rather than the life cycle emissions (L316-324).
Q7: All abbreviations used in tables and figures should be defined in the table notes or figure captions.
A: Thanks, We defined the meaning of “NGWP” in the table notes of Table 1 (L273)
Q8: The paper must be revised by a native English speaker.
A: The English in this document has been checked by at least two professional editors before submitting, both native speakers of English. For a certificate, please see: http://www.textcheck.com/certificate/PvPbBV
Reviewer 2 Report
The manuscript was well written, clear and concise. Conclusions are consistent with the objectives. A weakness is the heavy dependence on Chinese sources, but the authors have justified the limitations, and there is value in reporting TN rather than nitrate N.
Author Response
C: The manuscript was well written, clear and concise. Conclusions are consistent with the objectives. A weakness is the heavy dependence on Chinese sources, but the authors have justified the limitations, and there is value in reporting TN rather than nitrate N.
A: Thanks for comments
Round 2
Reviewer 1 Report
The authors have addressed all the comments and suggestions I made in the first review. I have no further suggestions.
Author Response
Thanks for comments